# Combination of Sodium Nitroprusside and Controlled Atmosphere Maintains Postharvest Quality of Chestnuts through Enhancement of Antioxidant Capacity

**DOI:** 10.3390/foods13050706

**Published:** 2024-02-26

**Authors:** Linging Pang, Yuqian Jiang, Lan Chen, Chongxiao Shao, Li Li, Xiaodong Wang, Xihong Li, Yanfang Pan

**Affiliations:** 1College of Biosystems Engineering and Food Science, Zhejiang University, Hangzhou 310058, China; panglingling2015@163.com (L.P.); lili1984@zju.edu.cn (L.L.); 2Tianjin Gasin-DH Preservation Technologies Co., Ltd., Tianjin 300300, China; jsdh_shaochx@163.com; 3State Key Laboratory of Food Nutrition and Safety, College of Food Science and Engineering, Tianjin University of Science and Technology, Tianjin 300457, China; jiangyuqian@tust.edu.cn (Y.J.); wangxd0501@163.com (X.W.); 4Shanxi Fruit Industry Cold Chain New Material Co., Ltd., Tongchuan 727100, China; chenlan890804@163.com; 5Institute of Food Science and Technology, Chinese Academic of Agricultural Sciences, Beijing 100193, China

**Keywords:** combined treatment, physio-chemical properties, antioxidant substance, preservation quality

## Abstract

The purpose of this study was to clarify the effect of CA (controlled atmosphere, 2–3% O_2_ + 3% CO_2_) and NO (nitric oxide, generated by 0.4 nM sodium nitroprusside), alone or combined (CA + NO), on the physio-chemical properties, enzyme activities and antioxidant capacities of chestnuts during storage at 0 °C for 180 d. Compared with control (CT), CA and CA+NO both improved the storage quality of the samples, but only CA resulted in more ethanol production. Moreover, these improvements were further enhanced and ethanol synthesis was inhibited by the addition of NO. A spectrometer was used to assess the production of phenolic content (TPC) and activities of phenylalanine ammonia-lyase (PAL), superoxide dismutas (SOD), peroxidase (POD), catalase (CAT) and polyphenol oxidase (PPO) as influenced by CA or CA+NO treatments. Higher TPC, PAL, SOD, POD, CAT, and lower PPO were observed in CA alone, and more so in the combination with NO group. The increased antioxidant production and enhanced antioxidant activities contributed to scavenging reactive oxygen species (ROS) and reducing malondialdehyde (MDA). This study unveiled the correlations and differences between the effects of CA and CA+NO on storage quality, providing valuable insights into postharvest preservation and suggesting that the combination (CA+NO) was more beneficial for quality maintenance in chestnuts.

## 1. Introduction

The chestnut (*Castanea* cv Mollissima) is a widely grown plant in the world and has received widespread attention for its rich nutrients, unique flavor and bioactive substances [1]. Its worldwide production in 2018 reached 2.35 million tons, with China contributing 83% to the total, which still showed explosive growth in recent years [2]. However, the shelf life of chestnuts is seriously limited due to high moisture and metabolic activity [3]. In addition, undesired changes, such as calcification, nutrient loss and rotting, easily occurred in chestnuts if there was no appropriate storage [4,5,6]. Generally, low temperature combined with other physical, biological, or chemical treatments has been researched for delaying quality deterioration in chestnuts, thereby prolonging the shelf life [7,8,9].

Postharvest preservation conditions, such as temperature, humidity, gas component and regulatory factors, obviously impacted the physiological metabolism of chestnuts during postharvest storage. Meanwhile, the increase in respiration rate, degradation of starch, decrease in total phenolic content (TPC), changes in antioxidant enzyme activity, and accumulation of reactive oxygen species (ROS) determined the storage quality and commodity value of chestnuts [10,11,12]. In reports, water curing treatment [13], methyl jasmonate immersion [14], edible coating [7], dill seed essential oil treatment [15], and water curing combined with controlled atmosphere treatment [16] have been developed to maintain the visual, structural and metabolic qualities of chestnuts.

Controlled atmosphere storage, indicating a low oxygen (O_2_) and high carbon dioxide (CO_2_) atmosphere, is a common, easy-to-operate and effective method in maintaining the quality and extending the shelf life of postharvest fruits and vegetables [17]. Most postharvest characteristics in fruits, such as pigment transformation, water loss, reactive oxygen species and malondialdehyde (MDA) accumulation, were highly associated with respiration intensity and bioactive compound degradation, which was confirmed by the investigation of CA [17,18,19,20]. Nevertheless, long-term controlled atmosphere storage could cause anaerobic respiration and increase the amount of acetaldehyde and ethanol in fruits, which not only brought unpleasant odors but also was susceptible to inducing reactive oxygen species production [21,22]. Therefore, to alleviate acetaldehyde and ethanol synthesis, additional treatment needs to be combined with CA. Nitric oxide, as a signaling molecule, has been widely and effectively used in postharvest preservation of food to inhibit respiration metabolism or maintain cell membrane integrity [23,24,25,26]. The inhibition effects of NO on the respiration metabolism of apples [23] and pears [25] have been studied. In addition, several studies demonstrated that NO could block ethylene synthesis, thus resulting in a reduced respiration rate [27,28], and could inhibit ethanol production in winter jujube by reducing alcohol dehydrogenase (ADH) activity [29]. NO also acted on the antioxidant system of fruits and restrained the accumulation of ROS [30], further protecting cell membranes against oxidant damage. Moreover, positive effects of NO on phenolic compound accumulation have been reported [31,32,33], which was also beneficial to free radical scavenging [34], in addition to the ability to scavenge antioxidant enzymes [35]. Additionally, to improve the convenience and usability of NO, sodium nitroprusside (SNP) has been widely used as an NO donor to investigate the role of NO in postharvest storage of peaches [24], pears [30] and blueberries [36]. Based on this, the synergetic effects of CA and NO might be expected to further improve the quality and shelf life of chestnuts.

As far as we knew, most previous studies focused on the effects of either CA or NO alone on preservation quality; however, few studies investigated the synergistic effects of CA and NO on chestnuts. In this study, the physio-chemical properties and antioxidant capacity in postharvest chestnuts during storage were investigated in response to CT and CA alone and combined with NO. Specifically, the physio-chemical properties included respiration rate, color, weight loss, decay rate, and microbial count, as well as starch and soluble sugar content and amylase activity. The antioxidant capacity was reflected in changes in antioxidant substances (phenolics), activities of enzymes related to phenolic synthesis and antioxidants, and ROS scavenging capacity. In addition, the correlations and differences in the above characteristics between the effects of CT, CA and CA+NO were also elaborated. The combination compensated for the ethanol content production resulting from long-term CA storage.

## 2. Materials and Methods

### 2.1. Chestnuts, Treatment, and Sample Collection

Chestnuts (*Castanea* cv Mollissima) were harvested at commercial maturity in the early morning in August 2022, in Qianxi County, Hebei Province, China. Chestnuts were packed in commercial cartons and then transported to Tianjin University of Science and Technology in cold chain within 3 h. Chestnuts were precooled at 0 °C at 90–92% relative humidity (RH) for 24 h. Chestnuts with uniform size and ripeness, free of defects and diseases, were selected as experimental samples. Two-thirds of the sample was immersed in distilled water for 10 min at 0 °C, and the remaining third was immersed in 0.4 mM sodium nitroprusside (SNP) solution for 10 min at 0 °C. After air drying, chestnuts were separately packed in PE bags (250 mm × 300 mm, 0.03 mm thick, oxygen transmission rate 9447 mL·m^−^^2^·d^−^^1^) (200 g per bag) according to different pretreatments. Of the two-thirds of samples, one half was stored at 0 °C, 90–92% RH for 180 d, which was set as control group, recorded as CT. The other half was stored at 0 °C, 90–92% RH, combined with 2–3% O_2_ + 3% CO_2_ atmosphere for 180 d, which was set as controlled atmosphere group, recorded as CA. In the one-third of samples, all were stored at 0 °C, 90–92% RH, combined with 2–3% O_2_ + 3% CO_2_ for 180 d, which was set as controlled atmosphere and nitric oxide group, recorded as CA+NO. Samples were randomly collected at 30-day intervals (0, 30, 60, 90, 120, 150 and 180 d) to evaluate physiological changes in postharvest chestnuts.

### 2.2. Color Analysis

Color parameters, including L*, a* and b*, of two surfaces of chestnut kernels were measured using a colorimeter (HP-200, Hanpu Photoelectric Technology Co., Ltd., Shanghai, China). The a* and b* values could be expressed as hue angles (h°) [37]. The calculation is shown in Formula (1). Fifteen chestnuts from each group at each sampling point were randomly selected for one measurement. Every analysis was performed in triplicate and every test was repeated twice.
h° = arctan (b*/a*)(1)

### 2.3. Respiration Rate

Respiration rate of the chestnuts was determined using an infrared gas analyzer (Shandong Laiende Intelligent Technology Co., Ltd., Weifang, China), according to Guo et al. [38]. The concentrations of CO_2_ initially and after 30 min standing in a hermetic sample chamber were recorded. About 600 g of sample was used for each measurement. Every analysis was performed in triplicate and every test was repeated twice.

### 2.4. Ethanol Content

Ethanol content of the chestnuts was determined using the headspace gas chromatography method as described by Tian et al. [39]. Fifteen chestnut kernels were randomly selected, freeze-dried and ground into powder in liquid nitrogen for one measurement. About 100 g powder was dissolved with an equal amount of precooled trichloroacetic acid (20%), and homogenized in an ice water bath for 5 min. The homogenate (5 g) was put into a 10 mL headspace vial, incubated at 40 °C for 60 min. An amount of 0.5 mL gas sample was taken from the headspace vial using a gastight syringe, then injected into the gas chromatograph (Agilent 7890, Agilent Technologies, Santa Clara, CA, USA), which was equipped with a flame ionization detector to measure ethanol content. Every analysis was performed in triplicate and every test was repeated twice.

### 2.5. Weight Loss, Decay Rate and Microbial Count

The weight of the chestnuts was recorded at each sampling point. Weight loss was represented as the percentage of weight change relative to the initial. About 300 g sample was used for each measurement. Every analysis was performed in triplicate and every test was repeated twice.

Fifty samples from one bag (1 kg per bag) were randomly selected for decay rate assessment. Chestnuts with appearances of mildew, injury, brown or black spots were considered to be rotten [38]. Every analysis was performed in triplicate. Formula (2) was used, as follows.
Decay rate = Rotten numbers of chestnuts/50 × 100%(2)

Microbial count of the chestnuts was determined as described by Guo et al. [38]. Fifteen chestnuts from each group at each sampling point were randomly selected for each measurement. About 25 g of intact chestnuts was loaded into a sterile stomacher bag, containing 225 mL phosphate buffer solution. After homogenization for 2 min, the homogenate was diluted 10-fold. Aliquots (1 mL) of the dilution were spread on plate count agar plates, and microbial count was recorded after incubation at 30 °C for 28 h and expressed as logarithm of colony forming units per gram (log (CFU·g^−1^)). Every analysis was performed in triplicate and every test was repeated twice.

### 2.6. Starch Content, Soluble Sugar Content and Amylase Activity

Starch content, soluble sugar content and amylase activity of chestnuts were measured as described by Guo et al. [38]. Fifteen chestnut kernels were randomly selected, freeze-dried and ground into powder in liquid nitrogen for one measurement. Every analysis was performed in triplicate and every test was repeated twice. The standard curves of glucose, sucrose and maltose were established, for quantitative detection of starch, soluble sugar and amylase activity, respectively. Additionally, the color reagents of dinitrosalicylic acid (DNS) (3.15 g DNS, 2.5 mL crystalline phenol, 131 mL NaOH (2 mol·L^−1^) and 2.5 mL sodium sulfite dissolved in 500 mL aqueous solution containing 185 g potassium sodium tartrate) and anthrone (0.5 mL anthrone ethyl acetate and 5.0 mL concentrated sulfuric acid) were prepared and used for further measurement.

For detection of starch content, 1 g powder was extracted with 10 mL ethanol (80%) in an 80 °C water bath for 40 min, cooled to room temperature, then filtered to collect the residue. The residue was dissolved with 20 mL distilled water, then heated in boiling water for 15 min to gelatinize. After that, 2 mL perchloric acid solution (9.5 mol·^−1^) was added and stirred for 30 min to hydrolyze starch. Then, the mixture was filtered, and the filtrate was diluted to 100 mL with distilled water, which was used as starch extract. After heating in boiling water for 10 min, the absorbance of the mixture (2.0 mL starch extract and 1.5 mL DNS) was measured at 540 nm, using a spectrometer (Spectra Max 190, Molecular Devices Corporation America, California, CA, USA). Formula (3) was used as follows.
Starch content = (*m*’*V* × *N*)/(*V*_s1_ × *m* × 10^6^) × 0.9 × 100%(3)
where *m*’ is the glucose content, μg; *V* and *V*_s1_ are the total volume and sampling volume of starch extract, respectively, mL; *N* is the dilution factor; *m* is the sample mass, g. 0.9 is the conversion factor from glucose to starch.

For detection of soluble sugar content, 2 g powder was extracted with 6 mL ethanol (80%) in an 80 °C water bath for 40 min, centrifuged at 4 °C, 10,000× *g* for 10 min. The supernatant was diluted to 50 mL with 80% ethanol, which was used as soluble sugar extract. After heating in boiling water for 10 min, the absorbance of the mixture (0.1 mL soluble sugar extract, 0.9 mL distilled water and 3 mL anthrone reagent) was measured at 630 nm. Formula (4) was used as follows.
Soluble sugar content = (*m*’’*V* × *N*)/(*V*_s2_ × *m* × 10^6^) × 100% (4)
where *m*’’ is the source content, μg; *V*_s2_ is the sample volume of soluble sugar extract, mL.

For detection of amylase activity, 10 g powder was extracted with distilled water (10 mL) for 20 min at room temperature, centrifuged at 4 °C and 10,000× *g* for 20 min, and the supernatant was collected and used as enzyme extract. The mixture (1 mL enzyme extract, 2.0 mL DNS and 1.0 mL starch solution (10 g·L^−1^)) was incubated in boiling water for 20 min. After cooling to room temperature, the mixture was diluted with 16 mL distilled water, and absorbance at 540 nm was measured every 10 s for 6 min. One unit of enzyme activity was expressed as the amount of soluble protein required to increase in absorbance by 1 per minute (U·g^−1^).

### 2.7. Total Phenolic Content (TPC) and Soluble Quinone Content (SQC)

For detection of TPC and SQC, fifteen chestnut kernels were randomly selected, freeze-dried and ground into powder in liquid nitrogen for one measurement. Every analysis was performed in triplicate and every test was repeated twice. An amount of 5 g of powder was dissolved with 25 mL methanol and left in darkness at 4 °C for 12 h. After centrifugation at 4 °C and 10,000× *g* for 10 min, the supernatant was collected and used as phenolic extract and quinone extract.

TPC was measured using the Folin–Ciocalteu method as described by Jiang et al. [40]. Amounts of 25 μL supernatant and 125 μL Folin reagent (Shanghai Yuanye Bio-Technology Co., Ltd., Shanghai, China) were added into each well of a 96-well plate and incubated for 10 min at 25 °C. Then, 125 μL Na_2_CO_3_ solution (7.5%) was added to each well and reacted for 30 min at 25 °C. TPC was determined by measuring absorbance at 765 nm using a spectrometer and the result expressed as g, gallic acid equivalent kg^−1^.

SQC was determined by measuring the absorbance of the above-collected supernatant at 437 nm and the result was expressed as A437.

### 2.8. Enzymatic Activities of Phenylalanine Ammonia-Lyase (PAL), Polyphenol Oxidase (PPO), Superoxide Dismutas (SOD), Peroxidase (POD) and Catalase (CAT)

For detection of enzyme activities, fifteen chestnut kernels were randomly selected, freeze-dried and ground into powder in liquid nitrogen for one measurement. Every analysis was performed in triplicate and every test was repeated twice. One unit of enzyme activity was expressed as the amount of soluble protein required to increase absorbance by 1 per minute (U·g^−1^).

PAL activity detection followed Assis et al. [41]. An amount of 5 g powder was homogenized with precooled 80% acetone (1:10 *w*/*v*) and extracted in a freezer for 15 min. After filtering and vacuum drying, the obtained acetone powder (0.5 g) was homogenized with 5 mL sodium borate buffer solution (100 mM, PH 8.8), containing 5 mM β-mercaptoethanol, 2 mM EDTA and 4% polyvinylpyrrolidone at 4 °C for 1 h, then centrifuged at 10,000× *g*, 4 °C to collect the supernatant, which was used as protein extract. Finally, 0.5 mL enzymatic extraction, 3 mL boric acid buffer (50 mM, PH 8.8) and 0.5 mL L-phenylalanine solution (20 mM) were mixed, incubated at 37 °C for 1 h, and measured for absorbance at 470 nm every 10 s for 6 min.

PPO and POD activity detection followed Jiang et al. [40]. An amount of 3 g powder was extracted with 3 mL phosphate buffer solution (100 mM, PH 5.5, 4% polyvinylpyrrolidone and 1% Triton X-100) in an ice bath. After centrifugation at 4 °C and 10,000× *g* for 30 min, the supernatant was collected for further analysis. To assay PPO activity, 10 μL supernatant, 200 μL phosphate buffer sodium (50 mM, PH 5.5) and 50 μL catechol (50 mM) were mixed, and absorbance was detected at 420 nm every 10 s for 6 min. To assay POD activity, 30 μL supernatant, 180 μL guaiacol (25 mM) and 30 μL H_2_O_2_ (500 mM) were mixed, and absorbance was detected at 470 nm every 10 s for 6 min.

CAT activity detection followed Li et al. [42]. An amount of 5 g powder was extracted with 5 mL phosphate buffer solution (100 mM, PH 7.5), containing 5 mM dithiothreitol and 5% polyvinylpyrrolidone, in an ice bath. After centrifugation at 4 °C and 10,000× *g* for 30 min, the supernatant (0.5 mL) and 2.9 mL H_2_O_2_ (20 mmol) was mixed, and absorbance was detected at 240 nm every 10 s for 6 min.

SOD activity was detected using the total superoxide dismutase assay kit (Shanghai Yuanye Bio-Technology Co., Ltd., Shanghai, China).

### 2.9. Hydrogen Peroxide (H_2_O_2_) Content, Superoxide Radical (O_2_^•−^) Content, 1,1-Diphenyl-2-Picrylhydrazyl (DPPH) Scavenging Ability and Malondialdehyde (MDA) Content

H_2_O_2_ content and O_2_^•−^ content were detected using the hydrogen peroxide assay kit (Beijing Solarbio Science and Technology Co. Ltd., Beijing, China) and superoxide anion assay kit (Shanghai Yuanye Bio-Technology Co., Ltd., Shanghai, China), respectively. The results were expressed as mmol·kg^−1^.

DPPH scavenging ability detection followed Tang et al. [43]. An amount of 25 μL phenolic extract was mixed with 175 μL DPPH solution (350 μM in methanol), and placed in darkness at 25 °C for 4 h. The results were recorded by measuring absorbance at 517 nm and expressed as mmol Trolox equivalent kg^−1^.

MDA content detection followed Jiang et al. [40]. An amount of 5 g powder was extracted with 5 mL trichloroacetic acid (100 g·L^−1^) and centrifuged at 10,000× *g*, 4 °C for 10 min. The supernatant (2 mL) was mixed with 2 mL thiobarbituric acid (6.7 g·L^−1^) and then incubated in boiling water for 20 min. After cooling to room temperature, the mixture’s absorbance was recorded at 450 nm, 532 nm and 600 nm. The result was expressed as µmol·kg^−1^.

### 2.10. Statistical Analyses

ANOVA was carried out using SPSS software 19.0 (SPSS Inc., Chicago, IL, USA). All values were expressed as mean ± standard deviation. Significance analysis of data was performed using Student’s *t*-test. Graphs were drawn with Origin 2021 software (OriginLab, Northampton, MA, USA).

## 3. Results

### 3.1. Physiological Characteristics (Color, Respiration Rate, Ethanol Content, Weight Loss, Decay Rate and Microbial Count)

The evaluation of physiological attributes, such as color, respiration rate, ethanol content, weight loss and microbial count, is an integral part of maintaining fruit quality and storage life during postharvest storage, directly influencing its nutritional and economic value. The L* and h° of chestnuts in CT, CA and CA+NO groups all showed downward trends during the whole storage period (Figure 1A,B). By comparing different groups, it could be seen that the L* of CT was significantly lower than that of treated chestnuts from day 30 to day 180. However, there were no differences in the L* of chestnuts between the CA and CA+NO groups within 60 days. Since day 90, the L* of chestnuts from all groups all presented dramatic decreases; however, the values in the CA+NO group always maintained an obviously higher level than those in the CA alone and CT groups (Figure 1A). The h° in CA+NO was 5.0–16.1% and 1.5–10.5% higher than those in CT and the CA alone, respectively, during the whole storage period, indicating more effective and immediate effects of the combination treatment on fresh chestnuts (Figure 1B).

Respiration was one of the important and initial factors affecting fruit metabolism. As in Figure 1C, the respiration rates of the chestnuts in the CT, CA and CA+NO groups all first increased and then slightly decreased throughout the storage period. Comparison revealed that the value in CA+NO-treated chestnuts was significantly lower than that in CT during the entire period. An evidently lower respiration rate of chestnuts in CA was also obtained, in comparison with CA+NO, after the 60th day. Specifically, on day 180, the respiration rates of CA+NO-treated chestnuts were 13.4% and 9.5% lower than the values in CT and CA, respectively. Ethanol, as an anaerobic respiration product, also serves as a crucial index for exploring changes in storage quality. As displayed in Figure 1D, the changing trend of ethanol content in each group was consistent with that of respiration rate. Generally, the value of CA-treated chestnuts was higher than those in CT and CA+NO. Significant differences were observed for chestnuts in CA compared with the CT and CA+NO groups over the whole storage period. On day 120, the ethanol content in all groups reached the maximum, where the value in CA was 18.6% and 54.9% higher than those in the CT and CA+NO groups, respectively.

Weight loss, decay rate and microbial count are essential quality attributes for postharvest fruit storage. For these characteristics, a trend toward improvement was found in CA and especially in CA+NO, and significant differences were exhibited among all the groups throughout the storage period (Figure 1E,F). At the end of storage, weight loss of the chestnuts in the CA and CA+NO groups was reduced by 11.2% and 24.2%, respectively, compared with CT (Figure 1E). Similar inhibition in the decay rate of chestnuts in the CA and CA+NO groups was also observed (Figure 1F). On day 180, the decay rate of samples in the CA and CA+NO groups decreased by 25.4% and 47.1%, respectively, compared with CT (Figure 1F). Consistent with decay rate, the microbial count of chestnuts in the CA and CA+NO groups was 13.8% and 24.7% lower than that in CT, at the final storage period (Figure 1G).

### 3.2. Starch Content, Soluble Sugar Content and Amylase Activity

As presented in Figure 2A, the starch content of all groups showed a downward trend throughout the storage period. Compared with CT, a significantly higher starch content was observed in chestnuts treated with CA+NO, while a higher value in the CA group was also observed, but not significantly. The soluble content of all samples showed a tendency of first increasing then decreasing after day 90 (Figure 2B). On day 90, the highest value in CA+NO treated chestnuts was 1.17-fold that in CT and CA. Although there were no obvious differences in soluble sugar content among each group, in the declining stage, the content in CA+NO was still 9.3% and 1.4% higher than those of CA and CT at the end of storage. Enzyme activity associated with starch degradation is shown in Figure 2C. Amylase activity in all groups increased in the first 30 d of storage, and decreased afterwards. Compared with CT, lower amylase activity was observed in treated groups during the entire storage period, especially in CA+NO. This was consistent with higher levels of starch in treated groups (CA, CA+NO) (Figure 2A) than in CT.

### 3.3. Total Phenolic Content (TPC), Soluble Quinone Content (SQC) and Activities of Phenylalanine Ammonia-Lyase (PAL), Polyphenol Oxidase, Superoxide Dismutas (SOD), Peroxidase (POD) and Catalase (CAT)

Polyphenols play an important role in the quality control of postharvest chestnuts due to their dual role as an antioxidant and a browning substrate. Therefore, combined with TPC, quality was evaluated from the perspective of browning products and enzyme activity in this study. As shown in Figure 3A, TPC of all groups increased within the first 90 d and then decreased until the end of storage, and it always remained higher in treated groups, especially CA+NO, compared to CT throughout the storage period. PAL activity in all samples increased continuously over time, and was 1.41-fold and 2.10-fold higher in CA and CA+NO groups, respectively, than in CT (Figure 3C). This explained the higher TPC in the treated groups from a biosynthetic perspective (Figure 3A). PPO activity in all groups presented a tendency towards first increasing and then decreasing after day 120 (Figure 3D). It could be clearly seen that there was an inverse relationship between PPO and TPC in all groups (Figure 3A,D). Additionally, POD activity in all samples also showed an increasing pattern followed by a decreasing trend (Figure 3F). Compared with CT, significant differences in POD activity were observed in CA-treated chestnuts from day 30 to day 150, and especially in CA+NO. The peak value in CA+NO (120 d) was 3.16-fold and 2.22-fold higher than those in CA (120 d) and CT (90 d), respectively. Similar trends in POD were observed in the tendencies of SOD and CAT (Figure 3E–G). Following the increase over the first 120 days, SOD activity in CA+NO was 1.24-fold and 1.52-fold higher than the highest value in CA (90 d) and CT (90 d), respectively (Figure 3E). In terms of CAT, the maximum activity was also observed in the CA+NO group on day 120, which was 1.19-fold and 1.28-fold higher than the maxima in CA (90 d) and CT (90 d), respectively (Figure 3G). As the enzymatic product, SQC of all groups kept growing throughout the storage period (Figure 3B). Closer observation revealed that CA and CA+NO treatments all significantly alleviated the production of SQC, compared with CT (Figure 3B). On day 180, SQC values in the CA and CA+NO groups were 32.7% and 41.1% lower than that of CT, respectively.

### 3.4. Hydrogen Peroxide (H_2_O_2_) Content, Superoxide Radical (O_2_^•−^)) Content, 1,1-Diphenyl-2-Picrylhydrazyl (DPPH) Scavenging Ability and Malondialdehyde (MDA) Content

As shown in Figure 4A, the H_2_O_2_ content in all groups gradually increased during the storage period. In the first 90 days, H_2_O_2_ content was not significantly different between the two treatment groups (CA and CA+NO), but both were obviously lower than that in CT. Furthermore, after day 90 stronger inhibition of H_2_O_2_ accumulation was observed in CA+NO treatment, compared with CA alone. On day 180, the H_2_O_2_ level was 29.5% and 32.1% lower than those in CA and CT separately. Similar inhibition of O_2_^−^ accumulation was observed in samples treated with CA, especially after treatment with CA+NO (Figure 4B). On day 180, O_2_^−^ content of chestnuts in the CA+NO group was reduced by 11.7% and 71.7% compared with CA and CT. The changing trend for DPPH scavenging activity among the groups was opposite to that for H_2_O_2_ and O_2_^−^ content (Figure 4A–C). At the end of storage, the value of DPPH scavenging activity in the CA+NO group was 31.9% and 34.1% higher than those in CA and CT respectively. It is worth noticing that changes in DPPH scavenging activity were consistent with TPC (Figure 3A and Figure 4C). MDA in all groups exhibited an upward trend during the entire storage period, with its content in CT on day 180 being 18.4-fold higher than at the start. Compared with CT, significant inhibition of MDA accumulation was observed in the CA+NO group after day 30 and the CA group after day 60. At the end of storage, MDA contents in CA+NO and CA were 7.6-fold and 9.1-fold higher than at the start.

### 3.5. Correlation Analysis

To unveil the potential relationship among physical quality attributes, bioactive compounds and antioxidant activities of chestnuts during postharvest storage, Pearson’s correlation was used for analysis (Figure 5). Respiration rate was positively related to weight loss, decay rate, microbial count, ethanol content, soluble sugar content, SQC, MDA and H_2_O_2_, O_2_^−^ (0.68 < r^2^ < 0.88), and negatively correlated with L* and h° (r^2^ = −0.79, −0.83) and starch content (r^2^ = 0.93). TPC, as one of the effective bioactive compounds, was negatively correlated with MDA, H_2_O_2_, and O_2_^−^ (−0.12 < r^2^ < −0.24), but positively correlated with DPPH scavenging capacity (r^2^ = 0.91). Positive correlation coefficients between TPC and PAL (r^2^ = 0.85), POD (r^2^ = 0.75), SOD (r^2^ = 0.87) and CAT (r^2^ = 0.89) were also observed. In addition, H_2_O_2_ and O_2_^−^ content also showed negative relationships with POD, SOD and CAT (−0.0048 < r^2^ < −0.53), although they were minor.

## 4. Discussion

Respiration and ROS metabolism, as physiological metabolic activities, play a vital role in the regulation of ripening, aging and quality of fruits during postharvest storage [44,45]. Among these attributes related to the metabolism, color is one of the most intuitive characteristics and closely related to fruit quality and consumer acceptability [46]. Compared with CT, the higher L* and h° in treated groups, especially CA + NO (Figure 1A,B), indicated CA alone or combined with NO was beneficial for sustaining the sensory quality of chestnuts. Considering the role of ethylene as an enhancer of fruit respiration [47], controlling its production might be essential to ensure the storage quality of fruits [48,49]. In previous studies, a controlled atmosphere has been widely used in postharvest preservation, showing reduced ethylene accumulation [50,51] by blocking its biosynthesis and absorption (Figure 6) [52,53,54]. This explained the lower respiration rate for chestnuts in treated groups (Figure 1C), relative to CT. Similar results were also found in other fruits stored under controlled atmosphere, including lime [55], sweet cherry [56], apple [57] and jujube [58]. Moreover, a much lower respiration rate was observed in the CA+NO treatment group (Figure 1C), which might be due to the synergistic effect of CA and NO, since NO could also bind with intermediates in the ethylene synthesis pathway (Figure 6) [59]. It is well established that fruits are prone to unpleasant odors due to massive synthesis of ethanol when exposed to a static controlled atmosphere for long periods [60], an effect that could be enhanced as oxygen decreases [61]. Therefore, it was understandable that higher ethanol content was observed in chestnuts in the CA group compared with CT (Figure 1D). However, ethanol accumulation was inhibited in CA+NO group (Figure 1D), possibly because NO also functioned as an ethanol production inhibitor by reducing alcohol dehydrogenase activity [29]. It has been reported that respiration is one of the main reasons for weight loss in fruits during storage [62], which explained the lower weight loss in the treated groups compared with CT (Figure 1E), combined with changes in the respiration rate among groups (Figure 1C). In addition, fruits with a high respiration rate were susceptible to perishing [63], which was supported by a lower decay rate (Figure 1F) and lower microbial count (Figure 1G) in the treated groups compared with CT in this study.

As we know, starch degradation is one of the main sources of soluble sugars [64,65], a process catalyzed by amylase [66,67]. Therefore, the higher starch content in the CA groups, especially CA+NO (Figure 2A), might be attributed to the corresponding lower amylase activity (Figure 2C). The lower respiration rate in treated chestnuts (Figure 1C) might be another reason for the maintenance of starch content, due to the role of soluble sugars as substrates in respiration metabolism [68,69]. Additionally, previous studies confirmed that starch was one of the main elements responsible for the maintenance of fruit cell structure [70,71,72,73], thereby improving postharvest storage quality, results that were consistent with changes in physical characteristics (Figure 1) among groups in this study. Similar results were observed in other plants stored under CA, such as sorghum [74] and ball peach [75]. There were also studies showing that NO treatment could downregulate genes related to starch degradation in cucumber [76] and kiwifruit [72], and lowered respiration rates in strawberry [77] and apple [23]. Furthermore, inhibition of amylase activity and respiration and maintenance of starch content were also observed in NO-treated banana [78] and pear [30,79]. The above findings in previous studies strongly supported our analysis of NO regulation of starch metabolism in this study (Figure 2). It was worth noting that soluble sugar first increased and then decreased (Figure 2C), unlike the changing trends in starch content and amylase activity, which might result from different intensities between the consumption (respiration rate) and production (starch degradation) of soluble sugar at different storage times [80].

Previous studies have shown that chestnut is rich in phenolics [81] and PAL is the first enzyme in the biosynthesis pathway [82]. Therefore, the higher activity of PAL in treated groups, especially in CA+NO (Figure 3C), might be one of the reasons for the correspondingly higher TPC compared with CT (Figure 3A). As shown in Figure 6, TPC was also significantly affected by PPO activity, which could catalyze the oxidation of phenolics to quinones [83,84]. Hence, the lower PPO activity in treated groups might be another reason for the corresponding higher TPC compared with CT (Figure 3D). In addition, the lower SQC in treated groups also echoed changes in TPC (Figure 3A,B). As shown in Figure 6, during the ROS scavenging process, O_2_^−^ was catalyzed by SOD to H_2_O_2_ [85], which was subsequently converted to O_2_ and H_2_O by CAT and POD [86]. In this study, the higher activities of SOD, CAT and POD (Figure 3E–G), combined with lower contents of H_2_O_2_ and O_2_^−^ (Figure 4A,B), were consistent with the relationship between antioxidant enzymes and ROS in previous studies [35,85,86]. Similar results were also found in apples [87] and bananas [88] stored under CA+NO. From a non-enzymatic perspective, high levels of TPC could improve scavenging properties against H_2_O_2_ and O_2_^−^ [89]. DPPH was always used to assess the antioxidant capacity of phenolics [90], which explained the consistency of TPC (Figure 3A) and DPPH scavenging activities (Figure 4C) among groups during the storage period. Moreover, consistent with changes in ROS (Figure 4A,B), CA and especially CA+NO treatment both inhibited MDA production, compared with CT (Figure 4D), which was probably due to the strong ROS scavenging effect in the treated groups, thereby slowing lipid oxidation [91] or cellular damage [92]. The results confirmed that the introduction of NO further improved the storage quality of chestnuts. Similar results were reported for plums [93], strawberries and lettuces [94], apples [87], bananas [88], and grapes [95] under CA+NO storage.

## 5. Conclusions

The results of this study showed that the addition of NO might be a simple and effective method for further improvement of chestnut quality during storage on the basis of CA treatment, compared with CA alone and CT. A lower respiration rate, higher L* and h°, less weight loss, and lower decay rate and microbial count, which are the basic physical features of storage quality maintenance, were observed in the combined treatment group. Moreover, reduced starch degradation and rich soluble sugar in the CA+NO treatment resulted from the restrained amylase activity and reduced respiration rate. CA+NO treatment also promoted ROS scavenging and slowed MDA accumulation by inhibiting antioxidant substances (phenolics) and regulating the activity of enzymes related to phenolic synthesis and antioxidants, such as PAL, PPO, POD, SOD and CAT. Additionally, ethanol production was also alleviated by the combination treatment; thus, the storage flavor and quality improved. The convenience and effectiveness support CA combined with NO as a practical approach to maintain preservation quality during storage of chestnuts and provide a strategy for postharvest preservation of other fruits or vegetables.

## Figures and Tables

**Figure 1 foods-13-00706-f001:**
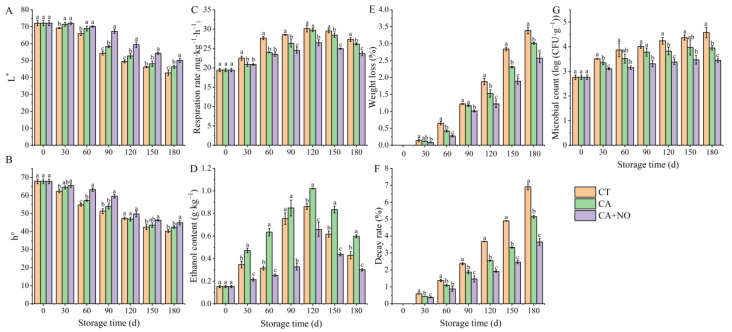
Effects of CA and CA+NO on L* (**A**), h° (**B**), respiration rate (**C**), ethanol content (**D**), weight loss (**E**), decay rate (**F**) and microbial count (**G**) of chestnuts during 180 days of storage at 0 °C. Different letters indicate significant differences between CT, CA and CA+NO within the same time point (*p* < 0.05).

**Figure 2 foods-13-00706-f002:**
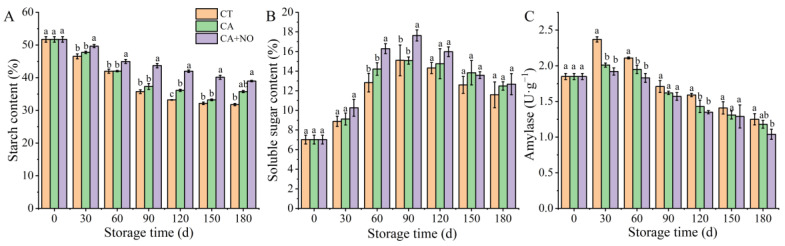
Effects of CA and CA+NO on starch content (**A**), soluble sugar content (**B**) and amylase activity (**C**) of chestnuts during 180 days of storage at 0 °C. Different letters indicate significant differences between CT, CA and CA+NO within the same time point (*p* < 0.05).

**Figure 3 foods-13-00706-f003:**
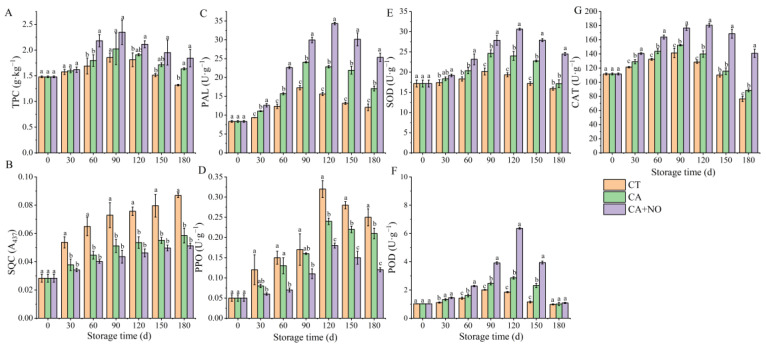
Effects of CA and CA+NO on TPC (**A**), SQC (**B**), PAL (**C**), PPO (**D**), SOD (**E**), POD (**F**) and CAT (**G**) of chestnuts during 180 days of storage at 0 °C. Different letters indicate significant differences between CT, CA and CA+NO within the same time point (*p* < 0.05).

**Figure 4 foods-13-00706-f004:**
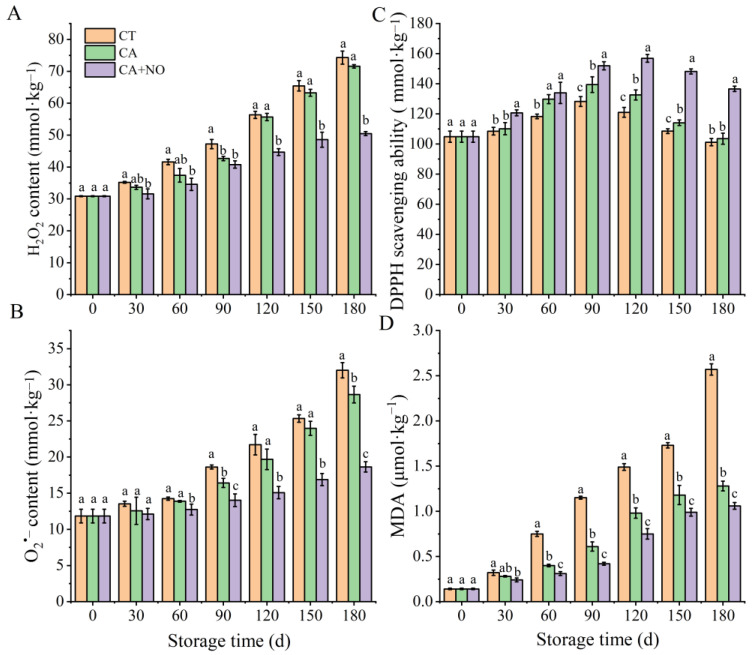
Effects of CA, CA+NO on H_2_O_2_ content (**A**), O_2_^•−^ content (**B**), DPPH scavenging ability (**C**) and MDA content (**D**) of chestnuts during 180 days of storage at 0 °C. Different letters indicate significant differences between CT, CA, CA+NO within the same time point (*p* < 0.05).

**Figure 5 foods-13-00706-f005:**
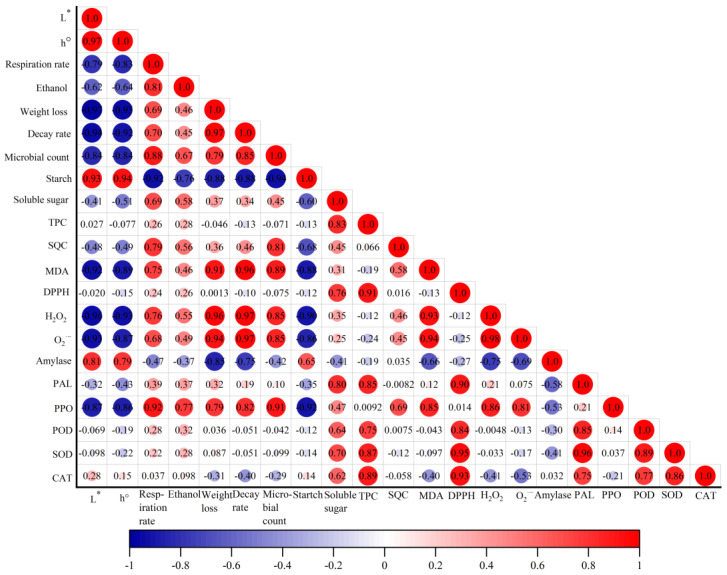
Pearson’s correlation coefficients for quality attributes of chestnuts, including L*, h°, respiration rate, weight loss, decay rate, microbial count, ethanol content, starch content, soluble sugar content, TPC, SQC, MDA, DPPH scavenging ability, H_2_O_2_, O_2_^−^ and PAL, PPO, SOD, POD, CAT and amylase. The red and blue colored dots are positive and negative correlations respectively, and the number presented is the correlation coefficient.

**Figure 6 foods-13-00706-f006:**
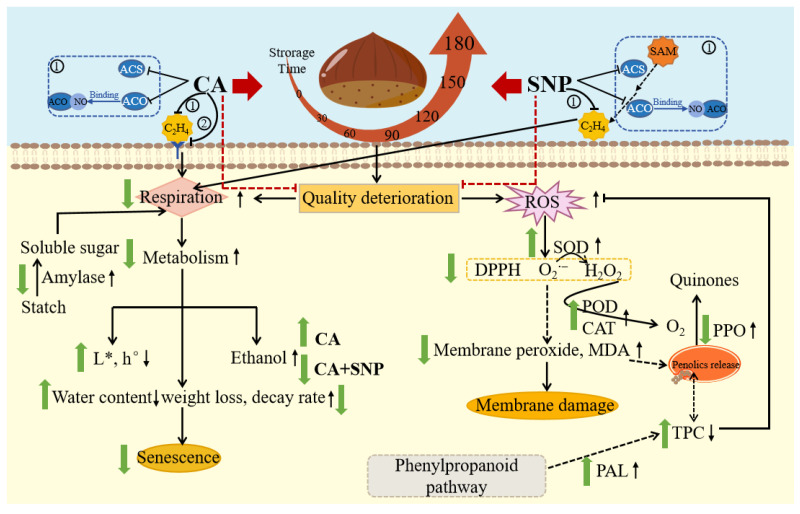
Schematic overview of regulation mechanism of CA alone or in combination with NO for storage quality in chestnuts. Green arrow indicates up- or downregulation in response to CA alone or in combination with NO. Black arrow indicates up- or downregulation in response to CT.

## Data Availability

The original contributions presented in the study are included in the article, further inquiries can be directed to the corresponding authors.

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
