# Peer review of "Combination of Sodium Nitroprusside and Controlled Atmosphere Maintains Postharvest Quality of Chestnuts through Enhancement of Antioxidant Capacity"

_foods, 2024, doi:10.3390/foods13050706_

Round 1

Reviewer 1 Report

Comments and Suggestions for Authors

General Comments:

            The title does not clearly reflect the research work. It is suggested to modify and harmonize with the objective.

            Modified or controlled atmosphere. Homogenize the term.

            The methodology seems correct to me, however, the sample size and repetitions are very low.

            The results, discussion and conclusions sections are adequate.

Abstract:

            It is missing to include or at least mention the most relevant materials and methods.

            Reduce results and focus on the most relevant findings. That is, what were the best treatments.

            Modify the key words. It is not recommended to include words contained in the title.

Introduction:

            Line 37: Add the scientific name.

            Line 39: What other countries grow chestnut trees?

            Line 43: How much are the losses due to not having adequate storage?

            Line 45: How much can the useful life be extended?

            Line 83-88: Improve the writing of the research objective.

Materials and Methods:

            Line 91: What variety of chestnut? What are their characteristics?

            Line 95: What size was the sample for each treatment?

            The size and characteristics of the samples are not clear. It should be clarified perfectly. It is handled differently in each section. It is important to specify why.

Results:

            The results are clear and well worked.

Discussion:

            It is a solid and well-argued discussion.

Conclusions:

            Make recommendations for future research related to the topic.

            Adjust the abstract and the objective according to the conclusions.

Comments on the Quality of English Language

The language is easily understood

Author Response

Dear reviewer:

Thank you very much for your report on our manuscript (foods-2873576). We greatly appreciate you for the constructive comments. The response  was listed below and in the attachment.

With best regards,

sincerely Yours

16 February, 2024

Reviewer 2 Report

Comments and Suggestions for Authors

The introduction briefly shows the advantages and the disadvantages of the modified atmosphere storage and the combinations of it with other treatments.

It is shown that the most important properties in the long-term storage of chestnut are the pigment transformations, water loss, degradation of bioactive compounds. The consequence of the anaerobic respiration such as increasing of the amount of alcohol and acetaldehyde which cause unpleasant odor and decrease the amount of bioactive compounds. The comments of the cited references show how the nitric oxide treatment can decrease these inconveniences and prolong the shelf-life time for more different fruits.

The methodological part show the experimental design, the used treatments during 180 days storage time, and the used methods such as color analysis, respiration rate measurement, detecting of the alcohol content, the weight loss and the microbiological contamination. The degradation of the active compounds shown by the evaluation of the starch content, soluble sugar content, amylase activity, total phenolic content (TPC) and soluble quinone content (SQC), enzymatic and antioxidant activity.

The results is shown by up-to-date statistical methods and programs and discussed with appropriate and relevant literature resources. The relationship between the analyzed parameters is shown by correlation analysis.

The conclusions are relevant and give information about the advantages of the combination of the controlled atmosphere and nitric oxide treatment such as improvement of the quality parameters of the chestnut during the long-term storage in comparation of the simple CA or other storage methods.

All of the figures are understandable and with good-enough quality, but the figure five is looks more like a table than a figure. May be it is all of the correction what I can suggest for the authors.

Author Response

(The authors gave the same response as above.)

Reviewer 3 Report

Comments and Suggestions for Authors

The manuscript presents a description of very comprehensive storage tests on chestnuts, the shelf life of which was extended by the use of a modified atmosphere and nitric oxide generated by sodium nitroprusside. The selection of characteristics of the post-harvest state of chestnuts and the methods of their examination raises no doubts. The authors clearly presented the research results and the impact of the applied storage treatments on the condition of chestnuts. The relationships between individual parameters were also presented and their relation to previously published literature data was discussed. 

The only shortcoming is the incorrectly labeled rows of quality attributes on the right side of the Pearson’s correlation coefficients graph in Figure 5. They are moved up one step, I think.

Apart from this single mistake, the entire article is a valuable compendium of knowledge on how to extend the shelf life of chestnuts using the proposed improvement of the MAP technique.

The manuscript may be published after minor corrections.

Author Response

(The authors gave the same response as above.)
